# Advances in Salt-Storage Materials, Road and Anti-Freezing Performances of Salt-Storage Asphalt Mixture: A Review

Yanhai Yang [1], Guanliang Chen [1], Ye Yang [1,2,*], Liang Yue [1,*] and Jian Xu [3]

1   School of Transportation and Geomatics Engineering, Shenyang Jianzhu University, Shenyang 110168, China
2   College of Transportation Engineering, Dalian Maritime University, Dalian 116026, China
3   Research Institute of Highway Ministry of Transport, Beijing 100088, China
*   Correspondence: yangye@sjzu.edu.cn (Y.Y.); yueliang@stu.sjzu.edu.cn (L.Y.)

**Abstract:** Salt storage asphalt pavement has been considered as a functional pavement that could effectively and actively melt snow on the road. Based on the previous studies, the macro melting snow and ice mechanism on the salt storage road is studied, high performance salt storage materials have been developed, as well as to analyze pavement and anti-freezing performance of salt storage asphalt mixture. Although some studies have evaluated salt storage asphalt mixtures and salt storage materials, there still remains many issues related to the slow-release effect of salt storage materials and road performance. Therefore, the article tries to review the key contents: mechanism for ice-snow melting of salt storage asphalt pavement, salt-storage materials design, salt-storage asphalt mixture-mix design. Additionally, key points concerning the road and anti-freezing performance of salt storage asphalt pavement were assessed. Finally, a series of important proposes for further investigations in this field have been presented.

**Keywords:** road engineering; salt storage asphalt pavement; salt storage material; performance; effect evaluation





## 1. Introduction

In the world, many countries suffer from serious snow and ice disasters. In snowy day, traffic accidents often occurred due to snow and ice. Meanwhile, highways and airports were closed to ensure traffic safety. The snow disaster caused great damage to transportation, and caused great losses to people's lives and property, industrial and agricultural production. Now, people have begun to realize that the severe cold weather and ice-snow roads have brought serious problems. Therefore, taking snow removal have been as an important task. At present, the main snow removal methods are divided into "passive" and "active" [1]. "Passive" measures mainly include manual snow removal, mechanical snow removal and chemical snow removal. "Active" measures mainly include thermal snow melting method, physical inhibition of freezing pavement, and salt storage asphalt pavement. However, many measures have some shortcomings, which even affect the service life of the pavement and surrounding environment [1–3]. Salt storage asphalt pavement has been recognized as an efficient, cost-effective, and environmentally friendly method to active snow removal.

As early as the 1960s, Switzerland and Germany first began to study the salt storage asphalt pavement, and developed a low freezing-point pavement with Verglimit antifreeze as an additive, which could reduce the freezing point of the pavement to −20 °C [4]. In the 1980s, the salt storage material Mafilon (MFL) was developed and produced by Japan, which is still used all over the world. MFL is added to the asphalt pavement instead of mineral powder, which could realize that the pavement is not easy to freeze at low temperature. In addition, the effective snow removal time of MFL reaches 6–10 years [5]. To date, Japan has laid a total surface area of over 4 million square meters of salt-storing asphalt pavement [6]. In 1995, Johannes developed the now widely used salt storage

material Verglimit-260 (V-260 for short) [7]. V-260 is made of $CaCl_2$ particles modified with hydrophobic agent. The product could replace the corresponding particle size aggregate in the mixture. The asphalt mixture with V-260 has good high and low temperature performance, poor water stability, but still meets the requirements. In 2001, Suren et al. used phenolic resin to wrap the meltable salt to prepare a new salt storage material [8]. The salt storage material could not be damaged without external force, and the external water could not enter the material. At the same time, the material has hydrophobicity, when the outside temperature is higher than 1 °C. When the temperature is lower than 1 °C, the salt storage material will become brittle and fragile, and the water can quickly penetrate into the structure to dissolve the salt.

China has gradually carried out the introduction and research of this technology since the salt storage asphalt pavement was laid on the LanShang expressway (Shaanxi) in 2009 [9]. Due to the technological monopoly of V-260 and MFL, numerous salt storage materials have emerged, as shown in Table 1. In 2018, Deicing and Thawy Material Used in Asphalt Mixture-Part 2: Salinization-based Material (JT/T 1210.2-2018) was came out [10].

**Table 1.** Domestic research and development of anti-icing product.

| Product | Researcher | Material Characteristics |
|---|---|---|
| Icebane | Liu [9] | slow-release |
| high elastic salt storage asphalt mixture | Yu [11] | high elastic self-stress deicing and low freezing-point |
| long-term salt storage filler | Ma [12] | useing pesticide coating technology |
| environmental protection snow melting coating material | Chen [13] | good snow melting and slow-release performance and greatly reduce the pollution of salt storage materials to the environment |
| anti-icing glutinous emulsified asphalt | Tan [14] | meet the needs of maintenance and active snow removal for in-service pavement |
| Anticoagulant ice microcapsule coating material | Chang [15] | used for the preventive maintenance of roads, with convenient and simple construction and high flexibility |

Due to the application of salt storage materials in asphalt pavement, it has the function of melting snow and ice while maintaining the original function and structural performance of the pavement, significantly improving the safety of the road in bad weather. However, the key components are mostly chloride, its precipitation of the formation of aqueous solution will induce a certain change in the performance of the mixture, and even damage part of the road performance, making it become the impact of the main factor in the promotion of the application. In addition, the type of material variety, physical parameters vary greatly, add the way and the impact on the snow and ice function is not uniform. In addition, the corresponding snow and ice effect evaluation, mostly using qualitative observation methods or self-developed equipment, has not yet formed a complete evaluation system and standards. Therefore, on the basis of systematic investigation of domestic and international research on the performance of salt storage asphalt mixture, this paper summarizes and analyze the mechanism of snow and ice suppression, evaluation methods, material types and application effects, and discuss in detail the impact of salt storage materials on the snow and ice removal function of the mixture and road performance. The existing problems and further research directions are pointed out, so as to provide reference for material selection and performance guarantee.

## 2. Formation of Pavement Icing and Mechanism of Ice-Snow Melting of Salt Storage Asphalt Pavement

### 2.1. Formation of Pavement Icing

The phenomenon of snow and ice on roads often occurs in low-temperature climate and large-area snowfall. The Institute of cryogenic science of Hokkaido University observed the ice and snow on the road surface, and proposed the snow on the roads into seven types:

new snow, snow powder, snow grain, snowboard, ice, ice film, and snow melt [16]. Under the action of various methods behaviors (vehicle loads, chemical treatment and thermal absorption), the snow type changes frequently on the road, and the change of snow state was shown in Figure 1.

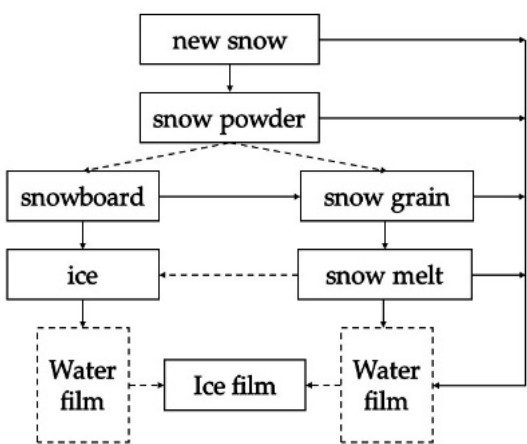

**Figure 1.** Patterns of snow cover change.

There is multi-layer snow structure, such as the upper snowboard, the lower ice block or the upper snow powder, and the lower ice film formed by wheel rolling. Road ice is mainly caused by the following. The snow is repeatedly rolled by the vehicle load, and the snow is gradually compacted to form snowboard, which begins to freeze and forms ice blocks. When the snow melts on the road during the day, ice begins to freeze-up at low temperature. Zhu summarized that when the temperature is between −10~0 °C, the relative humidity is greater than 75%, the wind speed is 0~10 m/s, and the wind direction is relatively fixed, the road is easy to freeze [17].

Therefore, the formation of pavement icing is caused by many factors, and the main influencing factors are pavement type, ponding and snow depth, meteorological conditions, pavement temperature, altitude, pavement structure depth and pavement slope [18,19].

*2.2. Mechanism for Ice-Snow Melting of Salt Storage Asphalt Pavement*

Salt storage asphalt pavement refers to the use of snow and ice melting materials to replace part of the minerals in the mixture, so that the salted material is stored in it [20]. During precipitation water (rainfall, snowfall, icing, freezing rain, etc.) and pavement structural voids [21]. The salinization in the mix can be actively released under the combined influence of the presence of release. In addition, through the pumping of vehicle loads, capillary pressure and concentration gradient and other effects, from the pavement structure inside to the road surface gradually precipitation [22]. Thus, reducing the freezing point of the road surface water solution, ice and snow and road adhesion between adhesion, to achieve the effect of snow melting. As the salt storage materials were added in the asphalt mixture and the limited freezing point of the components themselves as well as the slow release characteristics and slow release characteristics [23]. When a short period of snowfall or air temperature changes significantly, the de-icing effect of salt storage asphalt pavement will be affected. The salt solution layer still exists between the pavement and the snow and ice, which can significantly reduce the adhesion of the contact interface, and improve the efficiency of subsequent de-icing operation [24,25].

The microscopic mechanism of the adsorption and diffusion performance of salt storage components and deicing performance was studied. It was found that temperature, wind speed, pressure and concentration had a great impact on the adsorption and diffusion performance of salinity [22,23]. First of all, the freezing point of salt liquor is lower than water. Secondly, there are more solute molecules in salt liquor than aqueous liquor. According to the Raoult's law, adding different salts has the same effect on reducing the freezing point of the liquor. In addition, the reduction of the freezing point is proportional to the

molecular density of the liquor. Thirdly, the components of salt are very soluble in water. The existence of ions in salt liquor may damage the crystal network structure of water and prevent water from freezing [26,27]. Finally, the dissolution of salt is an exothermic process, which further promotes the snow melting [28]. Therefore, the salt storage liquor can effectively melt ice-snow on the pavement.

## 3. Salt Storage Materials

### 3.1. Classification and Application of Salt Storage Materials

Salt storage material refers to a composite material prepared by selecting a carrier to wrap the molten salt, and then processing the surface of the carrier wrapped with the molten salt [28]. According to different inclusions, salt storage materials could be divided into powder type, surface oil coating type and cement solidified type [24]. Powder type is used porous structural materials as the carrier, and meltable salt is wrapped and stored in the carrier after special processing. The representative product is MFL, as shown in Table 2 [29]. Physical properties of MFL are similar to mineral powder, and a certain amount of mineral powder can be replaced when building salt storage asphalt pavement. Salt gradually separates from the road surface under the action of vehicle load and osmotic pressure, which inhibits the formation of snow and ice on the road [30]. The surface oil coating type is to wrap the outer layer of the soluble salt with a layer of oily material to make it into granules. The representative product is V-260, as shown in Table 2 [29]. Due to the strong hydrophobicity of the surface wrapping material, the fusible salt in the salt storage material has a good slow-release performance after encountering water. When water penetrates into the interior of the salt storage material, the internal salt will dissolve in the water. At the same time, the wrapping material can still delay the contact between salt and water. Under the action of vehicle tire pressure and osmotic pressure, the salt liquor slowly diffuses from the interior of the road to achieve the effect of active ice and snow removal [29]. The cement solidified type uses cement to wrap the molten salt and crush it into composite materials with small particle size through crusher, which is represented by magnesite cement salt storage aggregate [30–32].

**Table 2.** Salt-storage material quality and technical index.

| Product | Item | Standards | Product | Item | Standards |
|---------|------|-----------|---------|------|-----------|
| MFL | Appearance Density/g·cm$^{-3}$ | Powder 2.25~2.35 | V-260 | Appearance Density/g·cm$^{-3}$ | Granularity 1.8 |
| | Grain Size/mm | <0.6 | | Grain Size/mm | 0.1~5 |
| | pH | 8~8.5 | | pH | 11~12 |
| | Water content/wt% | ≤0.5 | | Melting point/°C | 260 |
| | Salt content/wt% | 55 ± 10 | | Apparent density/g·ml$^{-1}$ | 0.84 |
| | Main Components | SiO$_2$, NaCl, MgO, CaCL$_2$, etc. | | - | - |

Self-developed snow melting, and ice suppression materials based on intellectual property rights and economic factors, domestic scholars have developed a number of snow and ice melting materials based on the design concepts of foreign products [30,33–37], as shown in Table 3. From its composition, the main components are still mainly chloride. From the main components of the composition is still mainly chloride, other types of salt. This is mainly due to the fact that chloride materials have the advantages of wide range, low price, stable performance and other characteristics; and the appearance of the product is mostly powder and granular. In addition, according to the promotion and application of the product It is found that most of the snow and ice melting materials developed in China are still in indoor trial research.

**Table 3.** Main components of salt-storage material.

| Product | Main Components | Grain Size/mm |
|---|---|---|
| ICB | Porous material, $SiO_2$, NaCl, MgO, $CaCL_2$, Coupling agent, Stabilizer, etc. | <0.6 |
| Salt storage zeolite aggregates | Natural zeolite, Cloride | 3~5, 5~10 |
| Environmentally friendly snow melting agent | Freezing point reduction main agent, Freezing delay component, Ice softening component, Stabilizer, etc. | <10 |
| Slow-release complex salt filler | Zeolite salt (dry zeolite powder, sodium chloride and tap water), Silicone resin, Solvent | <0.6 |
| Magnesia cement salt storage and slow-release material | Magnesia cement, Chloride, Aggregates, etc. | <3 |
| Snow melting and ice suppressing material RB-1 and RB-2 type | Carrier, $CaCL_2$ (RB-1 type) or ethylene glycol (RB-2 type), Hydrophobic agent, Coupling agent, Anti-corrosion agent | <0.15 |

Currently, salt storage materials typically have the problems of poor slow release performance and lack of long-term effectiveness. In order to solve the defects of salt storage materials, researchers have found that by coating the salt storage materials with polymers or adding surfactants [38,39]. The speed of salt diffusion from the film to the outside can be slowed down, the slow-release effect can be enhanced, and the purpose of long-term use can be achieved [11]. Adding surfactants wraps the snow melting salt to form a capsule structure. Therefore, the storage stability of the snow melting salt is improved, and the purpose of delaying the release of salt is achieved [40,41]. Among, the coating state of the coating directly affects the quality of the slow-release performance of the salt storage material. With the increase of the coating rate, the slow-release performance of the salt storage material will also be improved [42].

*3.2. Description of Salt Storage Material Performance*

Salt storage material is the key to the anti-freezing of salt storage pavement [43]. High performance salt storage materials should achieve the slow release, stability and prolong the service life of salt storage pavement. Many factors affect the performance of salt storage materials, including material composition, salt type, surface modifier, carrier, etc. [44,45]. Wu et al. found through experiments that adding surface modifiers can effectively adjust the slow-release performance of salt storage materials [45]. The high mass ratio of meltable salt could not effectively improve the ice-snow melting performance of salt storage asphalt pavement. On the contrary, the high mass ratio of meltable salt would reduce the road performance and durability, and increase the project cost. Surface modifiers could effectively enhance the slow-release performance of salt storage materials. The bearing capacity of the carrier to meltable salt is generally related to the abundance and adsorption performance of the material pores. The rich pore structure of the carrier material can store salt, and the high surface energy enables salt to be adsorbed on the surface of the carrier. Taking into account, both the anti-freezing and pavement performance, the optimal mix ratio of salt storage material should be determined reasonably.

Natural immersion analysis of the conductivity of solutions of salt storage materials generally assesses snowmelt capacity [43,45]. The Chinese standard JT/T 1210.2-2018 adopts the effective anion content test, which is mainly for the detection of chloride ions [15]. Due to slow-release characteristics, the test time of salt storage materials is often long. In order to shorten the test time, an accelerated immersion method was proposed. The dissolving-out tendency in the accelerated test was similar to that in the natural dissolving-out experiment [46]. There are many kinds of meltable salt, including chlorine salt, non-chlorine salt and composite salt. However, the current standards cannot quantitatively determine the performance of non-chloride and composite salt storage materials, and the industry standards need to be supplemented. Researchers also studied the hygroscopicity, skid resistance, environmental protection, recycle and other properties of filler type salt storage materials on asphalt pavement.

The deliquescence of salt ($CaCl_2$, $NaCl$, etc.) in salt storage materials makes them hygroscopic to a certain extent. In addition, the deliquescence of salt storage materials with $CaCl_2$ as the main component is more obvious than that of $NaCl$. After mixing the salt storage filler containing $CaCl_2$ into the mixture specimen and placing it for 24 h, it was found that a large amount of water appeared on the surface of the specimen [47]. After the snow frozen and melted, it was observed that the ordinary asphalt pavement dried rapidly, while the salt storage asphalt pavement containing $CaCl_2$ remained wet [48]. Therefore, the salt storage material will have a certain impact on the skid resistance of the pavement [6]. Therefore, when using salt storage materials that are mainly composed easy deliquescing salts ($CaCl_2$, $MgCl_2$, etc.), in order to ensure driving safety, the impact on the skid resistance of the pavement must be considered.

With the development of green transportation, researchers have begun to pay attention to the environmental protection of salt storage materials. It mainly studies the environmental impact of salt in salt storage asphalt pavement on nearby water quality, metal corrosion and so on. By analyzing the salt solution formed by rainwater flowing through the salt storage asphalt pavement, it was found that the chloride ion concentration in the salt solution did not reach the level that affected the environment [49]. The Chinese standard JT/T 973-2015 adopts carbon steel corrosion rate test to verify the corrosion of salt storage materials to metals [50]. According to the actual engineering experience of Beijing Daxing airport, it was found that the salt storage material had a weak impact on metal corrosion [51].

As for the recycling of salt storage asphalt pavement, the soluble salt in salt storage materials, especially $CaCl_2$, is not conducive to the recycling of asphalt mixture after grinding. Therefore, for the sake of recycling, many states in the United States do not recommend the use of salt storage materials containing $CaCl_2$ in asphalt pavement [52–55]. The relevant research on the recycling and utilization of salt storage asphalt mixture has not been carried out in China. Whether the salt storage pavement could be recycled still needs to be determined through research and demonstration.

## 4. Road Performance of Salt Storage Asphalt Mixture

### 4.1. Influence of Salt Storage Materials on Asphalt Mortar

At present, salt storage materials are mainly applied in two ways: (1) For the new pavement, it is added into the asphalt mixture instead of mineral powder or aggregate with the function of melting ice-snow and inhibiting freezing [56,57]; (2) For the in-service pavement, the salt storage asphalt pavement maintenance layer is prepared by adding modifiers, additives and salt storage materials with glutinous emulsified asphalt [58]. Filler type salt storage materials dominated by in research and application. Therefore, the paper uses salt storage materials in the form of fillers in the later study of the performance of salt storage asphalt mixture.

Asphalt mortar has a great impact on the road performance of asphalt mixture. Therefore, it is very important to clarify the influence of salt storage materials on asphalt mortar to explore the road performance of salt storage asphalt mixture. The different proportion of filler (or aggregate) in the mixture replaced by salt storage filler will influence on the properties of the formed asphalt mortar. Therefore, determining the proportion of salt storage filler is the prerequisite for the study of salt storage asphalt mortar. Xing et al. and Dou carried out 25 °C penetration test, 5 °C ductility test and the Brinell viscosity test of salt storage asphalt mortar [59,60]. The results were shown in Figure 2.

According to Figure 2a,b, the penetration and ductility of asphalt mortar gradually decreased with the increase of the replacement proportion of salt storage filler. The decrease of penetration indicated that the consistency and viscosity of asphalt mortar increased. The decrease of ductility indicated that the asphalt mortar became hard and brittle, which leaded to the low-temperature crack resistance decreased. The reason is that the salt storage filler and mineral powder have different adsorption, volume enhancement and physicochemical effects on asphalt. The salt storage filler has stronger adsorption on free

asphalt [61,62]. As the replacement proportion of salt storage filler increases, the structural asphalt increases, and the free asphalt decreases in salt storage asphalt mortar. Thus, the asphalt mortar hardens, the modulus increases.

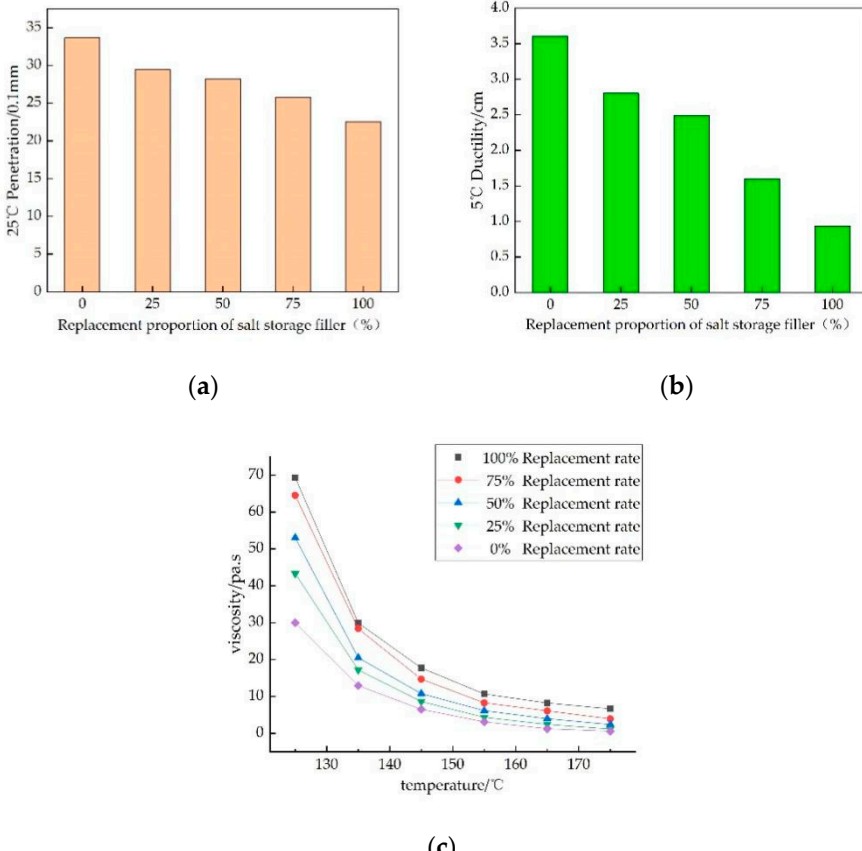

(**a**)　　　　　　　　　　　　　　　　　　　(**b**)

(**c**)

**Figure 2.** The performance of asphalt mortar varies with the amount of salt storage filler: (**a**) 25 °C penetration test; (**b**) 5 °C ductility test; (**c**) Brinell viscosity test.

Seen from Figure 2c, with the increase of temperature, the viscosity of salt storage asphalt mortar under each replacement rate was decreasing, but the rate of reduction was different. At 125~145 °C, asphalt mortar belonged to non-Newtonian fluid. In addition, the viscosity of salt storage asphalt mortar decreased rapidly with the increase of temperature at each replacement rate. At 145~175 °C, asphalt gradually changed to Newtonian fluid. Therefore, the viscosity of salinized asphalt mortar at each replacement rate decreased with the increase of temperature [59]. At the same time, the replacement rate of salt storage filler increased, the free asphalt in the asphalt mortar decreased, the internal friction of the mortar increased, and the viscosity of salt storage asphalt mortar increased.

*4.2. Mixture Design Composition*

According to the previous experience, powdered filler is often used to replace mineral powder about 5–7 wt.% in mixture. The surface is coated with salt filler to replace the fine aggregate in the mixture, and the addition amount accounts for about 5 wt.%. The particle size range of cement curing filler is about 2.36–13.2 mm, which is often used to replace the coarse and fine aggregates in asphalt mixture, with an addition of about 8 wt.% [27].

There are three replacement methods for salt storage asphalt mixture: equal volume replacement, equal mass replacement, and comprehensive optimization design, as shown in Figure 3 [6]. Equal mass replacement refers to the replacement of filler (or aggregate) of the same quality in the mixture with a certain quality of salt storage filler. Equal volume replacement refers to the replacement of the same volume of filler (or aggregate) in asphalt mixture with a certain volume of salt storage filler. Comprehensive optimization design

method is to mix the filler (or aggregate) with the salt storage filler in proportion, treat it as a new filler (or aggregate), and carry out the mix proportion design according to the Marshall design process [63].

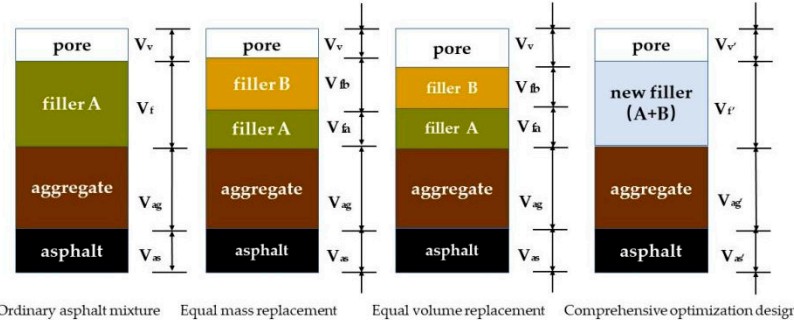

**Figure 3.** The comparison of salt-storage asphalt mixture mix design.

Equal quality replacement and equal volume replacement are simple to apply and do not modify the original grading design and production process. However, it will affect the road performance of the mixture and the effect of snow melting and ice suppression, and increase the difficulty of production and paving [64]. Comprehensive optimization design regards the mixing of different fillers as a new filler, recalculates its density, and designs the mix proportion. The performance of the designed mixture is better than equal quality replacement and equal volume replacement. In general, the salt storage asphalt mixture designed by the combination optimization design method is the best.

Climates, temperature, rainfall and snowfall have a great impact on the function of salt storage mixtures [65–67]. Therefore, in order to ensure the long-term function of the salt storage mixture, the mix proportion design needs to be optimized according to the zoning. The optimization is made up of three aspects (adaptability of different regions to adjust the void ratio of the mixture, appropriately changing the amount of salt storage filler, and adding additives) [68].

### 4.3. Comparison and Analysis of Road Performance

4.3.1. High Temperature Performance

In accordance with Chinese technical specification JTG F40-2004 [69], high temperature stability of asphalt mixtures with salt storage filler was evaluated by rutting test [70], which was expressed as dynamic stability (DS). According to the comparative studies [30,60,71], the influence of salt storage filler on AC-13 mixture was shown in Figure 4.

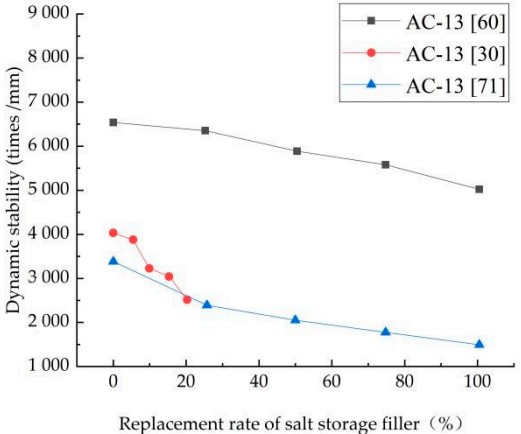

**Figure 4.** Rutting test of the same types of salt-storage mixture. (Shao 2015 [30], Dou 2015 [60], Sun 2011 [71]).

It could be seen from Figure 4 that the DS values decreased with the increase of salt storage filler content. It indicated that the addition of salt storage filler could reduce the high-temperature performance of asphalt mixtures. The salt particles and mineral materials are fully mixed and wrapped each other, which is a kind of heterogeneous powder material. The directly exposed salt particles reduce the interaction with asphalt, and could not form a relatively developed adsorption dissolution film on the surface of the salt. Next, the cohesion between salt particles decreases with the increase of salt content, the shear strength of asphalt mixture decreases, which could reduce the ability of asphalt mixtures to anti-rutting [71–73]. In combination with the relevant research, it was found that the high-temperature performance of different types of mixtures with salt storage filler was different, as shown in Figure 5 [62,63,68].

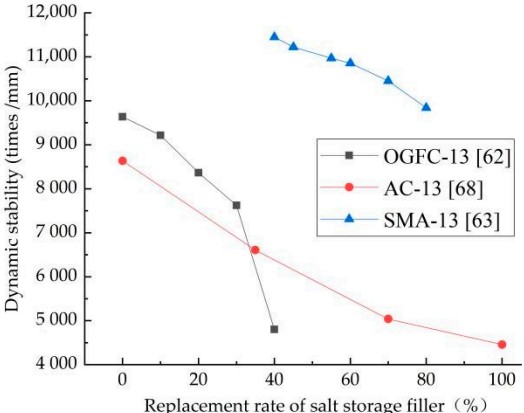

**Figure 5.** Rutting test of the different types of salt-storage mixtures. (Peng, et al. 2015 [62], Liu, et al. 2016 [63], Zhang, et al. 2011 [68]).

Figure 5 quantified that the DS of OGFC-13 with salt storage filler decreased significantly compared with AC-13 and SMA-13. The reason for difference is the free asphalt and the structural composition of mixture. After adding a small amount of salt storage filler, more free asphalt will appear. The structural composition characteristics of OGFC-13 are more sensitive to the cementation components in the mixtures, and the changes are more obvious. Therefore, it leads to OGFC-13 with salt storage filler more prone to deformation in rutting test.

### 4.3.2. Low Temperature Performance

The low-temperature crack resistance of the mixture with salt storage filler was evaluated using the ultimate flexural strain value ($\varepsilon_B$), measured by bending beam text at $-10 \pm 0.5\,°C$, as shown in Figure 6 [35,49,58].

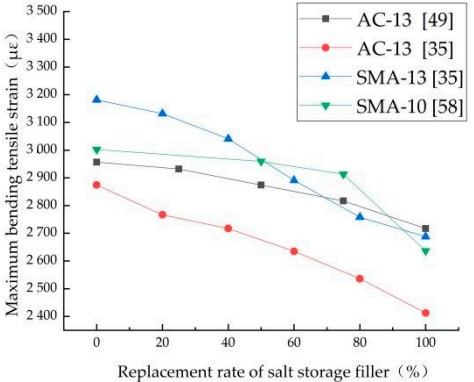

**Figure 6.** The low temperature cracking resistance of the salt-storage asphalt mixture. (Li, et al. 2012 [35], Stroup 2008 [49], Wang 2017 [32]).

As schematically illustrated in Figure 7, the $\varepsilon_B$ of salt storage asphalt mixture decreased with the increase of the replacement rate of salt compound. The addition of salt compounds reduced the low-temperature flexibility of asphalt mixture. The reason is that the total surface area of the filler gradually increases with the increase of the replacement rate of the salt storage filler. The viscoelasticity of the asphalt mortar decreases, and the brittleness increases [26,74]. Therefore, the low-temperature crack resistance of salt storage asphalt mixture is reduced.

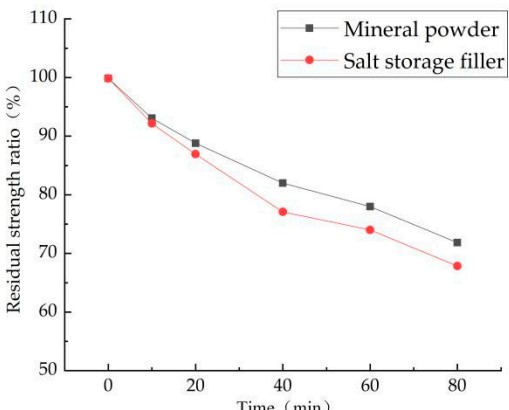

**Figure 7.** The scouring experiment of the asphalt mixture.

### 4.3.3. Water Stability Performance

Water stability is more important for salt storage asphalt pavement, especially in moisture and cold conditions. To explored water stability of salt storage asphalt pavement, using the dynamic water scouring and universal experimental instrument simulated the process of road surface water being pressed into the gap under positive pressure and sucked out under negative pressure under the repeated action of wheel rolling [36]. According to the freeze-thaw splitting test method, it measured the residual stability ratio (TSR) of AC-16 with salt storage filler, as shown in Figure 7.

From Figure 7, the TSR of salt storage mixture decreases with the increase of dynamic water scouring time. In addition, the reduction rate of ordinary asphalt mixture was less than salt storage mixture. It indicated that the water stability of salt storage pavement was lower than ordinary asphalt pavement. At the same time, wheel rolling and dynamic water scouring will aggravate the damage of salt storage asphalt pavement. In the salt storage asphalt mixture, deliquescence will occur at the filler, which greatly increases the number and porosity of the mixture and increases the contact area with water [75]. High-speed, high-pressure, hydrodynamic scouring and freeze-thaw will increase the size and number of pores, accelerate the salt dissolution, and thus aggravate the freeze-thaw damage [76,77]. Therefore, the addition of salt storage filler could weaken the water stability of asphalt mixture, and even could not meet the requirements of engineering, so it is necessary to add anti stripping agent [36].

In order to study water stability of different types of mixtures with the salt storage filler, the immersion Marshall test was carried out, the residual stability (RS) was measured, as shown in Figure 8 [36,61,72].

According to the data comparison in Figure 8, it was found that the RS decline of OGFC-13 with salt storage filler was more obvious, compared with AC-13 and SMA-13. It demonstrated that the water stability of OGFC-13 with salt storage filler was lower than AC-13 and SMA-13. The reason is that the change of volume index and the structural composition of mixture. There is a causal relationship between the change of volume index of salt storage asphalt mixture and the change of water stability performance [78]. During the immersion Marshall test, the 60 °C water bathing and 48 h bathing time accelerated the release of salt, resulting in poor adhesion between asphalt and aggregate. Therefore, it led to the loose internal structure of salt storage asphalt mixture and larger void ratio [79,80].

Due to the structural composition of OGFC-13, large characteristic void ratio makes easier for water to enter the interior of mixture. When water enters the interior of the salt storage asphalt mixture through the open pores, the salt contained in the salt storage filler is eroded and dissolved. In addition, then the asphalt mortar aggregate will form porous structure, resulting in more serious asphalt film peeling [77,81]. Therefore, water stability of OGFC-13 with salt storage filler declines more significantly than AC-13 and SMA-13.

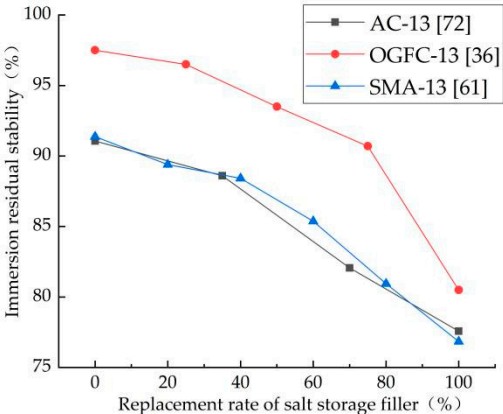

**Figure 8.** The immersion Marshall test of the different types of salt-storage mixtures. (Sun 2012 [36], Guo 2017 [61], Xu, et al. [72]).

*4.4. Road Performance Improvement Measures*

The addition of salt storage material in asphalt mixture damages some road performance to a certain extent, and will aggravate the road damage rate and reduce the road service life. Different improvement measures have been adopted to reduce the degradation of the performance of salt storage fillers in the mixture [82,83], as shown in Tables 4 and 5.

**Table 4.** Performance improvement of high elasticity modified asphalt on salt storage asphalt mixtures: HEA is high-elasticity asphalt; SBS is styrenic block copolymer.

| Evaluation Indicators | Mixture Type | | |
|---|---|---|---|
| | HEA + MFL | SBS + MFL | SBS |
| TSR (%) | 84.0 | 82.0 | 87.0 |
| Dynamic Stability (times/mm) | 9645 | 5096 | 8867 |
| Disruption strain ($\mu\varepsilon$) | 6194.4 | 2812.3 | 3769.5 |

**Table 5.** Performance improvement of Polyester fiber on salt storage asphalt mixtures.

| Evaluation Indicators | MFL | 0.3% Polyester Fiber + MFL |
|---|---|---|
| MS (%) | 78.0 | 85.0 |
| TSR (%) | 83.0 | 89.0 |
| Dynamic Stability (times/mm) | 2870 | 3560 |
| Disruption strain ($\mu\varepsilon$) | 2480 | 3080 |

It could be seen from Tables 4 and 5 that: (1) High elastic modified asphalt could significantly improve the properties of the binder, but the improvement of water stability was relatively small, and the TSR value was still lower than that of ordinary asphalt mixture.; (2) Adding polyester fiber can improve the road performance of salt-stored asphalt mixture to some extent, and the improvement of low temperature performance is obvious.

And Chen found that adding anti-stripping agent to the salt storage asphalt mixture, which could improve the water stability effectively [84]. In comparison, the effect of inorganic anti-peeling agent was better.

## 5. Snow and Ice Melting Performance and Timeliness

### 5.1. Snow Melting Performance

To explore the snow melting performance of salt storage asphalt mixture, the rut specimens were in the conditions of natural or artificial snowfall, compared the snow on the rut specimens. Then add digital image processing to quantitatively analyze the snow melting effect on the plate surface, and compare and evaluate the pavement exposure rate (the proportion of the area not covered by snow to the total area) [36], as shown in Figure 9.

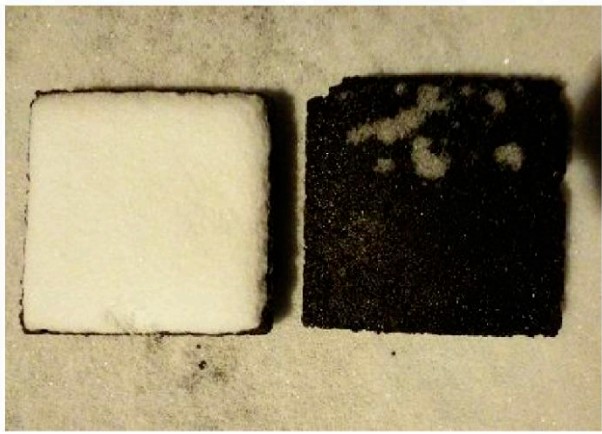

**Figure 9.** Melting snow text.

Through the comparison of the Figure 9, it was found that the snow layer of salt storage rut specimen was thin. In addition, the interface between rut specimen and snow was not frozen. However, the snow layer of rut specimen without salt storage filler was thick and evenly covered. This should be that the rut specimen with salt storage filler separated out salinity in the process of snowfall. It reduces the freezing point and the adhesion between the surface of rut specimen and ice and snow [85]. Comparing the snow melting ability of the specimens with different amounts of salt storage filler, the higher the proportion of salt storage filler is, the more obvious the snow melting effect is. Under different ambient temperatures, it was found that the effect of snow melting under the condition of $0 \sim -10\ °C$ is better than that under the condition of $-10 \sim -20\ °C$. At $-10 \sim -20\ °C$, although the salt storage asphalt pavement could not actively remove the snow on the surface, it could change the adhesion state between the snow and the road surface. It would be conducive to the further snow removal work. Therefore, it demonstrates that the salt storage asphalt pavement has good snow melting capacity

In order to determine the key factors affecting the snow melting performance, Han et al. obtained the grey entropy correlation degree of each factor by analyzing the conductivity (d) and correlation factors: water addition (L), void ratio (V), salt substitution (c), temperature (T) and dissolution time (t) [86]. That is to say, the significance of conductivity D and its associated factors (C, t, T, V and L) was quantified by grey correlation entropy analysis to obtain a quantitative characterization of this correlation [60,87]. The results are shown in Table 6.

**Table 6.** Grey entropy correlation degree of each correlation factor.

| Factors | L | V | c | T | t |
|---|---|---|---|---|---|
| Grey entropy correlation degree | 0.70455 | 0.54171 | 0.55288 | 0.59659 | 0.56382 |

It could be found from Table 6 that the grey entropy correlation degree of conductivity and influencing factors was sorted as follows: V < L < t < T < c. The replacement amount of salt storage filler is the key factor affecting the snow melting performance. In addition, temperature is the main factor affecting the snow melting performance. Therefore, on the

premise of meeting the road performance, the design of salt storage pavement should add a larger amount of salt filler.

### 5.2. Ice Melting Performance

For testing the ice melting performance of salt storage asphalt mixture, the Chinese standard JT/T 1210.2-2018 [15] proposed the ice melting rate test. It puts about 40 g ice cubes on the surface of standard Marshall specimens, puts them in a low-temperature environment box at −5 °C. After 2 h, the ice cubes are taken out and weighed the residual mass.

Rut test is also a common means to test the ice melting performance of salt storage asphalt mixture. Under the condition that the load action time, load size and temperature are certain, Wang made qualitative observation and description on the ice melting of salt storage asphalt mixture rut specimens [22]. According to different ice melting effects, the grade was made qualitative evaluation, shown in Table 7.

**Table 7.** Anti-freezing ability evaluation.

| Melting Grade | Surface Properties |
|---|---|
| excellent | The ice layer is broken and has no or weak adhesion with the surface, which is easy to peel off under the action of external force |
| good | The ice around the wheel is completely melted |
| medium | There is obvious moisture in the rolled part of the wheel compared with other parts |
| poor | The ice layer is closely bonded to the surface of the mixture, and there are only traces of wheels on the surface |

The ice melting ability of salt storage asphalt pavement is mainly manifested in reducing the bonding strength between pavement and ice surface [88]. To the evaluation index of the ice pavement bonding strength, the test bonding force was used, measured the ultimate tension between ice and road surface to characterize the bonding ability of ice and road surface, as shown in Figure 10 [74].

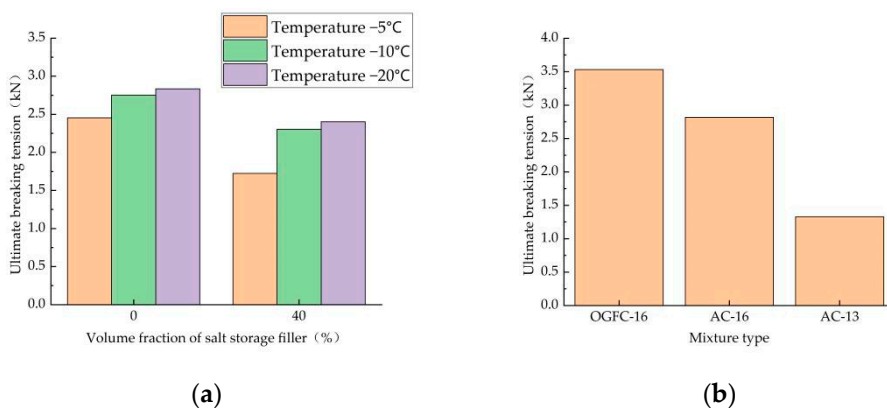

(**a**)　　　　　　　　　　　　　　　　　　(**b**)

**Figure 10.** Test of ice-pavement bonding strength: (**a**) test at different temperatures; (**b**) test with the different type mixtures.

According to the results of testing the ultimate breaking force in Figure 10a, it was found that under the same temperature, the ultimate breaking force of the salt storage mixture was less than the ordinary mixture. It indicated that the salt storage pavement could effectively reduce the bonding force between ice and the mixture. The ultimate breaking force of salt storage mixture at −5 °C was 27.8% lower than ordinary mixture, 11.7% lower at −10 °C and 10% lower at −20 °C. It indicated that the effect of salt storage filler decreased with the decrease of temperature. According to the results of Figure 10b,

OGFC-16 was 23% higher than AC-16. It illustrated that the grading type would affect the bonding force between the mixture and ice and the bonding force between open graded mixture and ice was greater than dense graded mixture. Comparing AC-16 and AC-13, it could be seen that the adhesion of AC-16 was 53% greater than that of AC-13. It illustrated that the nominal maximum particle size would also affect the adhesion between the mixture and ice. In addition, the adhesion increased with the increase of the nominal maximum particle size. Therefore, the ice pavement bonding strength is related to temperature, salt storage, nominal maximum particle size and type of mixture [88]. However, there is no unified test standard for the ice melting rate test. Therefore, it is suggested to supplement the qualitative evaluation index later.

*5.3. Service Life Evaluation of Snow and Ice Melting*

The salinity in the salt storage pavement will continue to lose in-service time. In addition, long-term snow and ice melting capacity will be decline continue. This also leads to the function life of the salt storage pavement to become shorter, which is inconsistent with service life. Due to no standard method to evaluate the functional life of salt storage pavement, the research on the functional life of salt storage pavement is relatively limited [89]. The detection capacity of salt storage pavement at home and abroad is determined by silver nitrate titration after several years of using. Silver nitrate titration can only be used for qualitative observation. In order to solve this problem, many researchers began to analyze and predict the long-term effectiveness of snow melting efficiency.

Natural dissolving-out test and accelerated dissolving-out test are commonly applied to predict the functional life of salt storage pavement. According to the cumulative release concentration of ions and the single release concentration of effective components, the release characteristics of effective components of salt storage asphalt mixture are characterized. Based on natural dissolving-out, Xu et al. found that the ion content and decay rate in salt storage asphalt mixture decreased with the increase of leaching time [72]. Theoretically, the salt in asphalt mixture will gradually precipitate due to the diffusion of chloride ions in the in-service time, until exhausted. Zheng based on the accelerated dissolving-out test and established a functional equation for predicting the functional life by using mathematical methods [36]. In addition, a model for predicting the functional life of salt storage pavement was proposed and shown as Equation (1).

$$T = e^{(100+b)/a}/K \tag{1}$$

where $T$ is the effective working time; $a$ and $b$ are fitting constant of accelerated test and natural test; $K$ is the area conversion factor.

The attenuation of the functional life of the salt storage pavement is caused by the release of the effective components in the salinized substances. The corresponding dissolution time, the replacement amount of the salinized substances, the void ratio, the temperature and the amount of water added will affect the snow melting life [32]. In order to determine the key factors affecting the function life of the salt storage pavement, Han et al. obtained the grey entropy correlation degree between the effective component residue and the influencing factors by analyzing the conductivity (d) and its related factors: water addition (L), void ratio (V), salt substitution (c), temperature (T) and dissolution time (t) [86]. It is that the importance of each factor is quantified by correlation analysis through the ash correlation entropy analysis of the residual amount of active ingredient $M_f$ and its correlation factors (C, t, T, V and L). A quantitative characterization of the magnitude of this correlation is obtained [60,87]. The gray correlation entropy analysis results are shown in Table 8.

**Table 8.** Grey entropy correlation between active component residual and influencing factors.

| Factors | L | V | c | T | t |
|---|---|---|---|---|---|
| Grey entropy correlation degree | 0.72714 | 0.81935 | 0.98520 | 0.77295 | 0.66644 |

It could be found from Table 8 that the grey entropy correlation degree between the residual amount of active ingredients and the influencing factors was sorted as follows: t < L < T < V < C. The salt substitution amount is the key factor affecting the function life. Temperature and void ratio are the main factors affecting the snow melting performance. With the increase of in-service time, the void ratio of salt storage asphalt mixture continues to increase, and the road performance shows a downward trend [63].

The gradation is also a factor affecting the functional life of salt storage pavement. Sun tested residual ion concentrations at different depths using a unidirectional seepage test method for salt storage mixtures of 3 typical grade combinations (AC-10, SMA, OGFC) with a test cycle of 21 d, as shown in Table 9 [90].

**Table 9.** Cumulative ion releasing process of different types of salt-storage mixtures.

| Depth (cm) | Residual Ion Concentration (mol/L) | | | | |
|---|---|---|---|---|---|
| | Primitive | 1 d | 7 d | 14 d | 21 d |
| AC-10 | | | | | |
| 0–1 | 0.3015 | 0.2843 | 0.2701 | 0.2310 | 0.2096 |
| 1–2 | 0.2978 | 0.2909 | 0.2742 | 0.2701 | 0.2469 |
| 2–3 | 0.2978 | 0.2964 | 0.2843 | 0.2714 | 0.2606 |
| 3–4 | 0.2992 | 0.2964 | 0.2909 | 0.2728 | 0.2767 |
| SMA | | | | | |
| 0–1 | 0.3606 | 0.3412 | 0.3141 | 0.2923 | 0.2576 |
| 1–2 | 0.3580 | 0.3553 | 0.3263 | 0.3053 | 0.2836 |
| 2–3 | 0.3597 | 0.3659 | 0.3474 | 0.3360 | 0.3255 |
| 3–4 | 0.3562 | 0.3483 | 0.3571 | 0.3289 | 0.3263 |
| OGFC | | | | | |
| 0–1 | 0.2604 | 0.2301 | 0.1798 | 0.1141 | 0.0891 |
| 1–2 | 0.2756 | 0.2631 | 0.2342 | 0.1844 | 0.1467 |
| 2–3 | 0.2798 | 0.2756 | 0.2604 | 0.2415 | 0.2199 |
| 3–4 | 0.2812 | 0.2825 | 0.2714 | 0.2466 | 0.2284 |

As could be seen from Table 9, residual ion concentrations in the surface layer (0–1 cm) and deep layer (3–4 cm) of three gradation type (AC-10, SMA, OGFC) decreased, respectively, by 30.47%, 28.56%, 65.8% and 7.52%, 8.39%, 18.76% in 21 d. It is proved that the gradation type has some influence on the salt release ability of salt storage asphalt mixture. Since water cannot seep down, the effect on the surface layer is only at the stage of immersion in still water. For better connected gradation types, such as OGFC, water mainly scours the mixture, resulting in the release of the active ingredient at a relatively faster and higher concentration. For the gradation type with interconnected pores, the active component in the deep layer decreases more quickly because the water can be immersed into the deep layer more easily. However, because of the slow process of water seepage, the release process of the active ingredient in the deep layer is slower with the increase of cycle, which can ensure the continuity and slow release of the active ingredient.

Yang selected three typical gradation (AC-13, SMA-13, OGFC-13) asphalt mixtures without salt storage filler to conduct multiple adhesion tests at −5 °C, −10 °C and −15 °C [91]. The average value of adhesion measured for many times was taken as the evaluation standard of failure. Carried out continuous adhesion test on salt storage asphalt mixtures of different grading types, melted the ice in the interface, and measured the precipitation concentration of effective components, as shown in Table 10. Completely natural soaking was used to test the concentration of effective composition analysis, as shown in Figure 11. By comparing the effective concentration and single release concentration of the three mixtures, the optimal effective release time was obtained. Since the test conditions corresponded to the actual rainstorm conditions, the local rainfall conditions

could be converted into rainstorm weather according to the rainfall. Therefore, the function life of the salt storage pavement could be calculated.

**Table 10.** The critical concentration of the mixture corresponds to the bonding force.

| Mixture Type | Cohesive Force (Kn) | Critical Concentration (mol/L) |
|---|---|---|
| AC-13 | 1.810 | 0.000625 |
| SMA-13 | 1.553 | 0.00137 |
| OGFC-13 | 1.369 | 0.0024 |

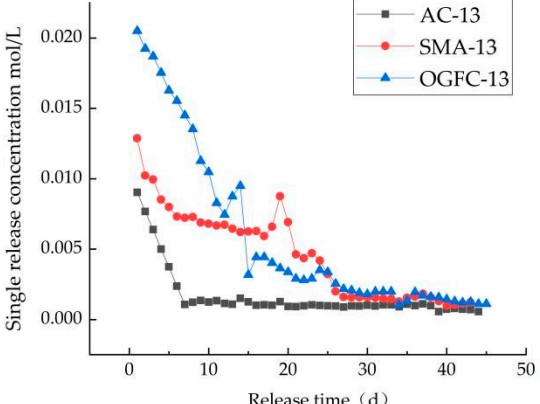

**Figure 11.** Single release concentration of ions under different types of salt-storage mixtures.

Combining Table 4 and Figure 11, it could be seen that the critical concentration of AC-13 was 0.000625 mol/L, the corresponding release time was 39 days and the converted effective period was 2.6 years. The critical concentration of SMA-13 was 0.00137 mol/L, the corresponding release time was 39 days, and the converted effective period was 2.3 years. The critical concentration of OGFC-13 was 0.0024 mol/L, the corresponding release time was 2.6 days and the converted effective period was 1.73 years, which was far less than the functional life of the other two asphalt mixtures [91].

Therefore, in order to ensure the functional life of the salt storage pavement, the salt storage material with good slow-release performance is preferred. The design void ratio can be appropriately reduced in the mix design of the salt storage asphalt mixture. During the construction, the full compaction of the asphalt mixture should be ensured. It is recommended to pave the salt storage pavement in autumn and winter [92,93]. It can not only ensure the anti-freezing performance in the first year, but also extend the functional life.

## 6. Conclusions and Outlook

Based on the research on the mechanism of salt storage pavement, the road performance of salt storage materials, salt storage asphalt mixture and the function of snow melting and ice suppression, the authors draw the following conclusions and suggestions for further research:

(1) Through macroscopic observation, it is found that the precipitation and migration of salt under the coupling action of mechanics and chemistry make the salt storage asphalt pavement realize active snow melting and ice melting. However, the quantitative analysis of local salt precipitation, diffusion and migration remains to be further studied. The key point of prolonging the service and functional life of salt storage asphalt pavement is to select appropriate meltable salt, hydrophobic agent and carrier to prepare high-performance salt storage materials. The slow-release performance of salt storage materials is generally not ideal. It is suggested that future research should be carried out in the control of slow release, environmental response and

corrosion protection of salt storage materials, so as to achieve long-term snow melting and ice suppression. The differences and accuracy of different evaluation methods are compared and analyzed in order to put forward reasonable test conditions and evaluation criteria.

(2) The design method of salt storage asphalt mixture is based on Marshall design theory, but there are still some problems in practical application. Compared with ordinary asphalt mixture, the high-temperature performance, low-temperature performance and water stability of salt storage asphalt mixture are lower. The water stability decreases the most obviously. The road performance of different types of salt storage asphalt mixtures is also different. The SMA mixture with salt storage filler has the best road performance and OGFC mixture with salt storage filler is the worst. Reasons for salt storage materials impacting on asphalt pavement: the reduction of adhesion of asphalt mortar after salt dissolution and the change of structural proportion of asphalt mortar. However, the deterioration mechanism of salt storage asphalt pavement is still lack. It is suggested that different kinds of materials should be selected, and suitable improvement measures should be taken to ensure the durability of the mixture, taking into account road performance, ease of use, and economic factors. Theoretical research on road performance of salt-storage asphalt mixture especially the multi-scale structural damage and performance degradation model under multi-field coupling is strengthened.

(3) From the experiments and engineering, the effect of snow and ice melting of salt storage asphalt pavement is good. In practical projects, the good effective function life of salt storage asphalt pavement is generally 2–3 years. There is still residual salt release in the later stage, but the effect of snow melting and ice suppression is not ideal. Different types of salt storage asphalt pavement have different functional life. Compared with OGFC salt storage asphalt pavement, AC and SMA salt storage asphalt pavement can ensure more stable and lasting salt release and longer functional life. The appropriate replacement rate, temperature and void ratio of salt storage materials are the key to realize the long functional life of salt storage asphalt pavement. It is recommended that in the future there be more accurate means of quantitatively evaluating the snow and ice melting effect of salt storage pavements, as well as establishing more accurate functional life prediction models.

**Author Contributions:** Conceptualization, G.C. and Y.Y (Yanhai Yang); methodology, G.C.; validation, Y.Y. (Ye Yang) and L.Y.; formal analysis, G.C.; data curation, G.C. and Y.Y. (Ye Yang); writing—review and editing, G.C. and Y.Y. (Ye Yang); supervision, Y.Y. (Yanhai Yang) and J.X. All authors have read and agreed to the published version of the manuscript.

**Funding:** This study was supported by the Liaoning Distinguished Professor Program grant NO. tpjs2017003.

**Institutional Review Board Statement:** Not applicable.

**Informed Consent Statement:** Not applicable.

**Data Availability Statement:** Not applicable.

**Acknowledgments:** This research was performed at Shenyang Jianzhu University.

**Conflicts of Interest:** The authors declare no conflict of interest.

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
