# Peer review of "Advances in Salt-Storage Materials, Road and Anti-Freezing Performances of Salt-Storage Asphalt Mixture: A Review"

_coatings, doi:10.3390/coatings12091323_

Round 1
Reviewer 1 Report
The article entitled.'' advances in salt storage materials, road and anti-freezing performances of salt storage asphalt mixture, a review'' briefly presented detailed information about salt storage materials and the role of salt storage asphalt pavement in the anti-freezing mechanisms, especially on roads. I like and recommend this article for publication after a few minor changes.
1. Detailed comparison of different salt-storage materials is not added in the introduction.
2. Objectives are not clear and explained. Please add 2-3 lines to explain why this MS is needed to be published. Research gaps should be highlighted, and novelty or contribution to the field must be added in a separate paragraph under a new heading.
3. Insufficient references in the MS. Need to add a few latest (2021-2022) references.
4. Figures' captions must be explained.
5. Line 545 ''Author draws following conclusions, or authors draw following conclusions?. Correct it.
6. Journal names should be abbreviated.
7. Several grammatical and sentence errors are here. Please proofread the MS.
6. d
raw
s the following conclusions
Author Response
Point 1: Detailed comparison of different salt-storage materials is not added in the introduction.
Response 1: Thank you for the reviewer. The comparison of different salt storage materials was presented in tables containing the properties of the materials, as shown in Tables 2 and 3.
Table 2. Salt-storage material quality and technical index
Product |
Item |
Standards |
Product |
Item |
Standards |
MFL |
Appearance |
Powder |
V-260 |
Appearance |
Granularity |
Density/g.cm-3 |
2.25~2.35 |
Density/g.cm-3 |
1.8 |
||
Grain Size /mm |
< 0.6 |
Grain Size /mm |
0.1~5 |
||
pH |
8~8.5 |
pH |
11~12 |
||
Water content /wt% |
≤0.5 |
Melting point/°C |
260 |
||
Salt content/wt% |
55±10 |
Apparent density/g.ml-1 |
0.84 |
||
Main Components |
SiO2, NaCl, MgO, CaCL2, etc. |
|
|
Table 3. Domestic research and development of salt-storage material
Product |
Main Components |
Grain Size /mm |
ICB |
Porous material, SiO2, NaCl, MgO, CaCL2, Coupling agent, Stabilizer, etc. |
< 0.6 |
Salt storage zeolite aggregates |
Natural zeolite, Cloride |
3~5, 5~10 |
Environmentally friendly snow melting agent |
Freezing point reduction main agent, Freezing delay component, Ice softening component, Stabilizer, etc. |
< 10 |
Slow-release complex salt filler |
Zeolite salt (dry zeolite powder, sodium chloride and tap water), Silicone resin, Solvent |
< 0.6 |
Magnesia cement salt storage and slow-release material |
Magnesia cement, Chloride, Aggregates, etc. |
< 3 |
Snow melting and ice suppressing material RB-1 and RB-2 type |
Carrier, CaCL2 (RB-1 type) or ethylene glycol (RB-2 type), Hydrophobic agent, Coupling agent, Anti-corrosion agent |
< 0.15 |
Point 2: Objectives are not clear and explained. Please add 2-3 lines to explain why this MS is needed to be published. Research gaps should be highlighted, and novelty or contribution to the field must be added in a separate paragraph under a new heading.
Response 2: Because of the application of salt storage materials in asphalt pavement, it has the function of melting snow and ice while maintaining the original function and structural performance of the pavement, significantly improving the safety of the road in bad weather. But the key components are mostly chloride, its precipitation of the formation of aqueous solution will induce a certain change in the performance of the mixture, and even damage part of the road performance, making it become the impact of the main factor in the promotion of the application. In addition, the type of material variety, physical parameters vary greatly, add the way and the impact on the snow and ice function is not uniform. And the corresponding snow and ice effect evaluation, mostly using qualitative observation methods or self-developed equipment, has not yet formed a complete evaluation system and standards. Therefore, on the basis of systematic investigation of domestic and international research on the performance of salt storage asphalt mixture, this paper summarize and analyze the mechanism of snow and ice suppression, evaluation methods, material types and application effects, and discuss in detail the impact of salt storage materials on the snow and ice removal function of the mixture and road performance. The existing problems and further research directions are pointed out, so as to provide reference for material selection and performance guarantee. (line 65-82)
Point 3: Insufficient references in the MS. Need to add a few latest (2021-2022) references.
Response 3: Thank you for the reviewer. Thank you for the reviewer. The author team improved the content of the literature and cited additional literature.
- Chang R, Wang J, Qin Y C, et al. Study on Synthesis and Service Properties of Anticoagulant Ice Microcapsule Coating Material. Mater. Sci. Eng, 2021, 2021,7423113, doi.org/10.1155/2021/7423113.
- Kang J. The Research on Anticoagulant Mixture of Ice.Master’s thesis, Chognqing Jiaotong University,China,
- Huang W,Yang H Q,Cong Y F,et al. Research on the Performance of the De-ice Materials use in Asphalt Pavement.Applied Chemical Industry, 2016,45,1215-1219, org/10.16581/j.cnki.issn1671-3206.20160427.029.
- Federico A, Longo S, Felice G. Phyto-based Sodium Chloride Hydrogel for Highway Winter Maintenance of Porous Asphalt Pavements. Build. Mater, 2022, 319, 126082,doi.org/10.1016/j.conbuildmat.2021.126082.
- Sajid H U, Naik D L, Kiran R. Improving the Ice-melting Capacity of Traditional Deicers. Build. Mater, 2021, 271, 121527, doi.org/10.1016/j.conbuildmat.2020.121527.
- Wang Z H, Guo L J. Application of Anticoagulant Ice Microcapsule Material in Micro-surfacing Technology. International Conference on Applied Mechanics and Structural Materials,2021, org/10.1088/1742-6596/2206/1/012024.
- Zhang Y L, Wang X S, Meng H L. Research on Self-melting Snow Material of Asphalt Pavement. International Conference on Advances in Modern Physics Sciences and Engineering Technology,2021, org/10.1088/1742-6596/2242/1/012013 .
- Guo Pe, Cai D Y, Liu J. Evaluation of Long-term Performance of Salt Storage Anti-icing Asphalt Mixture. Science Technology and Engineering,2022,22,4136-4142.
- Wang M M, Gao H R, Chai MM, Study on the Performances of an Anticoagulant Ice Coating Material for Asphalt Pavement. China, International Conference on Advanced Materials and Ecological Environment, AMEE, 2021, org/10.1088/1742-6596/2168/1/012014.
- Fu L, Zhou H, Yuan P,et al. Damage fracture characterization of asphalt mixtures considering freeze-thaw cycling and aging effects based on acoustic emission m Materials.2021, 14, 5930, doi.org/10.3390/ma14205930.
- Yang Y H, Wang H B, Yang Y. Characterization of Air Voids in Cold Recycled Mixtures with Emulsified Asphalt Under Freeze-Thaw Cycles. Rep, 2022,36,21110128, doi.org/10. 11896 /cldb. 21110128.
- Luo S,Yang X. Performance Evaluation of High-elastic Asphalt Mixture Containing Deicing Agent Mafilon. Build. Mater, 2015,94,494-501, doi.org/10.1016/j.conbuildmat.2015.07.064.
- Ma T, Geng L, Ding X H,et al. Experimental Study of Deicing Asphalt Mixture with Anti-icing Additives. Build. Mater, 2016,127,653-662, doi.org/10.1016/j.conbuildmat.2016.10.018.
- Chen Study on the Durability and Snowmelt Perdurability of Salt Anti-freezing Asphalt Pavement. Master’s thesis, Chang’an University, China, 2013.
- Li J M, Ji G Z, Kong E C, Liang G X. Analysis on Influence Factors of Fatigue Performance for Epoxy Asphalt Mixture Based on Grey Relation Degree Theory.Journal of East China Jiaotong University, 2017,34,14-20, org/10.16749/j.cnki.jecjtu.2017.01.003.
- Sun D H. Study on Typical Gradation and Residual Life Prediction Low Freezing Point Ultra-thin Wear Layers with Considering Anti-freezing Ice Ability. Master’s thesis, Harbin Institute of Technology, China, 2021.
Point 4:Figures' captions must be explained.
Response 4: Thank you for the reviewer. Numbers in parentheses represent the references from which this data was obtained in the images. It's marked in the Figure 5, 6, 7and 9.
Point 5:Line 545 ''Author draws following conclusions, or authors draw following conclusions?. Correct it.
Response 5: Thank you for the reviewer. Corrected to the authors draw the following conclusions. (line 595-596)
Point 6:Journal names should be abbreviated.
Response 6: Thank you for the reviewer. Since some of the journal names do not have abbreviated names, the author team has revised those journals whose abbreviated names can be found. (Cement Concrete Res, China J. Highw. Transp,J. China Univ. Min. Technol,J. Highw. Transp,etc.)
Point 7: Several grammatical and sentence errors are here. Please proofread the MS.
Response 7: Thank you for the reviewer.
- However, many measures have some shortcomings, which even affect the service life of the pavement and surrounding environment [1,2,3]. (line 36-37)
- To date, Japan has laid a total surface area of over 4 million square metres of salt-storing asphalt pavement [6].(line 47-48)
- Road ice is mainly caused by the following. (line 96-97)
- Currently, salt storage materials typically have the problems of poor slow release performance and lack of long-term effectiveness. (line 174-175)
- Natural immersion analysis of the conductivity of solutions of salt storage materials generally assesses snowmelt capacity [43,45]. (line 202-203)

Reviewer 2 Report
1. The research objectives and the significance of the research should be clarified in the manuscript.
2. There is a lack of citations for the reviews and the literature content should be improved as a Review paper.
3. The ice-snow melting of salt storage asphalt pavement is unclear. Figure 2 is not sufficient to express the mechanism, the figure should be revised by adding more arrows, and a heat map.
4. Classification and application of salt storage materials should be presented as flowcharts or tables containing the properties of materials.
5. When conducting a manuscript presenting various literature reviews, the author’s team should develop a Table that classifies the research method, research needs, research materials, and featured findings.
6. The author’s team should conduct systematically compare the advantages and disadvantageous of multiple solutions of salt-storage materials, then, suggest a solution for the drawbacks.
7. How to calculate the grey entropy correlation degree of each factor shown in Tables 1 and 3? The author’s team should briefly present the process.
8. Since the main objective of the research is to conduct a review on the performances of salt-storage asphalt mixtures, the main findings of the study only focus on a single source of citation number [31] as shown in Figure 12, Table 4, and Figure 13, containing AC-13, SMA-13, OGFC-13.
Therefore, it is suggested to diversify the results from more research.
9. The author’s team should present the research needs and suggestions for future studies on Advances in Salt-storage Materials, which problems should be focused on?

Author Response
Point 1: The research objectives and the significance of the research should be clarified in the manuscript.
Response 1: Thank you for the reviewer. Because of the application of salt storage materials in asphalt pavement, it has the function of melting snow and ice while maintaining the original function and structural performance of the pavement, significantly improving the safety of the road in bad weather. But the key components are mostly chloride, its precipitation of the formation of aqueous solution will induce a certain change in the performance of the mixture, and even damage part of the road performance, making it become the impact of the main factor in the promotion of the application. In addition, the type of material variety, physical parameters vary greatly, add the way and the impact on the snow and ice function is not uniform. And the corresponding snow and ice effect evaluation, mostly using qualitative observation methods or self-developed equipment, has not yet formed a complete evaluation system and standards. Therefore, on the basis of systematic investigation of domestic and international research on the performance of salt storage asphalt mixture, this paper summarize and analyze the mechanism of snow and ice suppression, evaluation methods, material types and application effects, and discuss in detail the impact of salt storage materials on the snow and ice removal function of the mixture and road performance. The existing problems and further research directions are pointed out, so as to provide reference for material selection and performance guarantee. (line 65-82)
Point 2: There is a lack of citations for the reviews and the literature content should be improved as a Review paper.
Response 2: Thank you for the reviewer. The author team improved the content of the literature and cited additional literature.
- Chang R, Wang J, Qin Y C, et al. Study on Synthesis and Service Properties of Anticoagulant Ice Microcapsule Coating Material. Mater. Sci. Eng, 2021, 2021,7423113, doi.org/10.1155/2021/7423113.
- Kang J. The Research on Anticoagulant Mixture of Ice.Master’s thesis, Chognqing Jiaotong University,China,
- Huang W,Yang H Q,Cong Y F,et al. Research on the Performance of the De-ice Materials use in Asphalt Pavement.Applied Chemical Industry, 2016,45,1215-1219, org/10.16581/j.cnki.issn1671-3206.20160427.029.
- Federico A, Longo S, Felice G. Phyto-based Sodium Chloride Hydrogel for Highway Winter Maintenance of Porous Asphalt Pavements. Build. Mater, 2022, 319, 126082,doi.org/10.1016/j.conbuildmat.2021.126082.
- Sajid H U, Naik D L, Kiran R. Improving the Ice-melting Capacity of Traditional Deicers. Build. Mater, 2021, 271, 121527, doi.org/10.1016/j.conbuildmat.2020.121527.
- Wang Z H, Guo L J. Application of Anticoagulant Ice Microcapsule Material in Micro-surfacing Technology. International Conference on Applied Mechanics and Structural Materials,2021, org/10.1088/1742-6596/2206/1/012024.
- Zhang Y L, Wang X S, Meng H L. Research on Self-melting Snow Material of Asphalt Pavement. International Conference on Advances in Modern Physics Sciences and Engineering Technology,2021, org/10.1088/1742-6596/2242/1/012013 .
- Guo Pe, Cai D Y, Liu J. Evaluation of Long-term Performance of Salt Storage Anti-icing Asphalt Mixture. Science Technology and Engineering,2022,22,4136-4142.
- Wang M M, Gao H R, Chai MM, Study on the Performances of an Anticoagulant Ice Coating Material for Asphalt Pavement. China, International Conference on Advanced Materials and Ecological Environment, AMEE, 2021, org/10.1088/1742-6596/2168/1/012014.
- Fu L, Zhou H, Yuan P,et al. Damage fracture characterization of asphalt mixtures considering freeze-thaw cycling and aging effects based on acoustic emission m Materials.2021, 14, 5930, doi.org/10.3390/ma14205930.
- Yang Y H, Wang H B, Yang Y. Characterization of Air Voids in Cold Recycled Mixtures with Emulsified Asphalt Under Freeze-Thaw Cycles. Rep, 2022,36,21110128, doi.org/10. 11896 /cldb. 21110128.
- Luo S,Yang X. Performance Evaluation of High-elastic Asphalt Mixture Containing Deicing Agent Mafilon. Build. Mater, 2015,94,494-501, doi.org/10.1016/j.conbuildmat.2015.07.064.
- Ma T, Geng L, Ding X H,et al. Experimental Study of Deicing Asphalt Mixture with Anti-icing Additives. Build. Mater, 2016,127,653-662, doi.org/10.1016/j.conbuildmat.2016.10.018.
- Chen Study on the Durability and Snowmelt Perdurability of Salt Anti-freezing Asphalt Pavement. Master’s thesis, Chang’an University, China, 2013.
- Li J M, Ji G Z, Kong E C, Liang G X. Analysis on Influence Factors of Fatigue Performance for Epoxy Asphalt Mixture Based on Grey Relation Degree Theory.Journal of East China Jiaotong University, 2017,34,14-20, org/10.16749/j.cnki.jecjtu.2017.01.003.
- Sun D H. Study on Typical Gradation and Residual Life Prediction Low Freezing Point Ultra-thin Wear Layers with Considering Anti-freezing Ice Ability. Master’s thesis, Harbin Institute of Technology, China, 2021.
Point 3: The ice-snow melting of salt storage asphalt pavement is unclear. Figure 2 is not sufficient to express the mechanism, the figure should be revised by adding more arrows, and a heat map.
Response 3: Thank you for the reviewer. Salt storage asphalt pavement refers to the use of snow and ice melting materials to replace part of the minerals in the mixture, so that the salted material is stored in it. During precipitation water (rainfall, snowfall, icing, freezing rain, etc.) and pavement structural voids. The salinization in the mix can be actively released under the combined influence of the presence of release. And through the pumping of vehicle loads, capillary pressure and concentration gradient and other effects, from the pavement structure inside to the road surface gradually precipitation, as shown in Figure 2. Thus reducing the freezing point of the road surface water solution, ice and snow and road adhesion between adhesion, to achieve the effect of snow melting. As the salt storage materials were added in the asphalt mixture and the limited freezing point of the components themselves as well as the slow release characteristics and slow release characteristics. When a short period of snowfall or air temperature changes significantly, the de-icing effect of salt storage asphalt pavement will be affected. The salt solution layer still exists between the pavement and the snow and ice, which can significantly reduce the adhesion of the contact interface, and improve the efficiency of subsequent de-icing operation. (line 108-123)
Point 4: Classification and application of salt storage materials should be presented as flowcharts or tables containing the properties of materials.
Response 4: The comparison of different salt storage materials was presented in tables containing the properties of the materials, as shown in tables 2 and 3.
Table 2. Salt-storage material quality and technical index
Product |
Item |
Standards |
Product |
Item |
Standards |
MFL |
Appearance |
Powder |
V-260 |
Appearance |
Granularity |
Density/g.cm-3 |
2.25~2.35 |
Density/g.cm-3 |
1.8 |
||
Grain Size /mm |
< 0.6 |
Grain Size /mm |
0.1~5 |
||
pH |
8~8.5 |
pH |
11~12 |
||
Water content /wt% |
≤0.5 |
Melting point/°C |
260 |
||
Salt content/wt% |
55±10 |
Apparent density/g.ml-1 |
0.84 |
||
Main Components |
SiO2, NaCl, MgO, CaCL2, etc. |
|
|
Table 3. Domestic research and development of salt-storage material
Product |
Main Components |
Grain Size /mm |
ICB |
Porous material, SiO2, NaCl, MgO, CaCL2, Coupling agent, Stabilizer, etc. |
< 0.6 |
Salt storage zeolite aggregates |
Natural zeolite, Cloride |
3~5, 5~10 |
Environmentally friendly snow melting agent |
Freezing point reduction main agent, Freezing delay component, Ice softening component, Stabilizer, etc. |
< 10 |
Slow-release complex salt filler |
Zeolite salt (dry zeolite powder, sodium chloride and tap water), Silicone resin, Solvent |
< 0.6 |
Magnesia cement salt storage and slow-release material |
Magnesia cement, Chloride, Aggregates, etc. |
< 3 |
Snow melting and ice suppressing material RB-1 and RB-2 type |
Carrier, CaCL2 (RB-1 type) or ethylene glycol (RB-2 type), Hydrophobic agent, Coupling agent, Anti-corrosion agent |
< 0.15 |
Point 5:When conducting a manuscript presenting various literature reviews, the author’s team should develop a Table that classifies the research method, research needs, research materials, and featured findings.
Response 5: Thank you for the reviewer. The author team has presented the research methodology, research needs, research materials and featured findings in a tabular format, as shown in Table 1.
Table 1. Domestic research and development of anti-icing product
product |
researcher |
material characteristics |
Icebane |
Liu [9] |
slow-release |
high elastic salt storage asphalt mixture |
Yu [10] |
high elastic self-stress deicing and low freezing-point |
long-term salt storage filler |
Ma [11] |
useing pesticide coating technology |
environmental protection snow melting coating material |
Chen [12] |
good snow melting and slow-release performance and greatly reduce the pollution of salt storage materials to the environment |
anti-icing glutinous emulsified asphalt |
Tan [13] |
meet the needs of maintenance and active snow removal for in-service pavement |
Anticoagulant ice microcapsule coating material |
Chang [14] |
used for the preventive maintenance of roads, with convenient and simple construction and high flexibility |
Point 6:The author’s team should conduct systematically compare the advantages and disadvantageous of multiple solutions of salt-storage materials, then, suggest a solution for the drawbacks.
Response 6: The addition of salt storage material in asphalt mixture damages some road performance to a certain extent, and will aggravate the road damage rate and reduce the road service life. Different improvement measures have been adopted to reduce the degradation of the performance of salt storage fillers in the mixture, as shown in Tables 4 to 5. (line 405-409)
Table 4 Performance improvement of high elasticity modified asphalt on salt storage asphalt mixtures: HEA is high-elasticity asphalt; SBS is styrenic block copolymer.
Evaluation Indicators |
Mixture type |
||
HEA+MFL |
SBS+MFL |
SBS |
|
TSR (%) |
84.0 |
82.0 |
87.0 |
Dynamic Stability (times/mm) |
9645 |
5096 |
8867 |
Disruption strain (με) |
6194.4 |
2812.3 |
3769.5 |
Table 5 Performance improvement of Polyester fiber on salt storage asphalt mixtures
Evaluation Indicators |
MFL |
0.3% Polyester fiber+MFL |
MS (%) |
78.0 |
85.0 |
TSR (%) |
83.0 |
89.0 |
Dynamic Stability (times/mm) |
2870 |
3560 |
Disruption strain (με) |
2480 |
3080 |
It could be seen from Table 3 to Table 4 that: (1) High elastic modified asphalt could significantly improve the properties of the binder, but the improvement of water stability was relatively small, and the TSR value was still lower than that of ordinary asphalt mixture.; (2) Adding polyester fiber can improve the road performance of salt-stored asphalt mixture to some extent, and the improvement of low temperature performance is obvious. (line 413-418)
And Chen found that adding anti-stripping agent to the salt storage asphalt mixture, which could improve the water stability effectively. In comparison, the effect of inorganic anti-peeling agent was better. (line 419-421)
Point 7: How to calculate the grey entropy correlation degree of each factor shown in Tables 1 and 3? The author’s team should briefly present the process.
Response 7: That is to say, the significance of conductivity D and its associated factors (C, t, T, V and L) was quantified by grey correlation entropy analysis to obtain a quantitative characterization of this correlation. (line 449-452)
It is that the importance of each factor is quantified by correlation analysis through the ash correlation entropy analysis of the residual amount of active ingredient Mf and its correlation factors ( C, t, T, V and L). A quantitative characterization of the magnitude of this correlation is obtained. (line 532-536)
Point 8:Since the main objective of the research is to conduct a review on the performances of salt-storage asphalt mixtures, the main findings of the study only focus on a single source of citation number [31] as shown in Figure 12, Table 4, and Figure 13, containing AC-13, SMA-13, OGFC-13.Therefore, it is suggested to diversify the results from more research.
Response 8: Thank you for the reviewer. Our team has found other studies to fill in and modify.
The gradation is also a factor affecting the functional life of salt storage pavement. Sun tested residual ion concentrations at different depths using a unidirectional seepage test method for salt storage mixtures of 3 typical grade combinations (AC-10, SMA, OGFC) with a test cycle of 21d, as shown in Table 9. (line 544-547)
Table 9. Cumulative ion releasing process of different types salt-storage mixtures
AC-10 |
|||||
Depth(cm) |
Residual ion concentration (mol/L) |
||||
primitive |
1d |
7d |
14d |
21d |
|
0-1 |
0.3015 |
0.2843 |
0.2701 |
0.2310 |
0.2096 |
1-2 |
0.2978 |
0.2909 |
0.2742 |
0.2701 |
0.2469 |
2-3 |
0.2978 |
0.2964 |
0.2843 |
0.2714 |
0.2606 |
3-4 |
0.2992 |
0.2964 |
0.2909 |
0.2728 |
0.2767 |
SMA |
|||||
Depth(cm)
|
Residual ion concentration (mol/L) |
||||
primitive |
1d |
7d |
14d |
21d |
|
0-1 |
0.3606 |
0.3412 |
0.3141 |
0.2923 |
0.2576 |
1-2 |
0.3580 |
0.3553 |
0.3263 |
0.3053 |
0.2836 |
2-3 |
0.3597 |
0.3659 |
0.3474 |
0.3360 |
0.3255 |
3-4 |
0.3562 |
0.3483 |
0.3571 |
0.3289 |
0.3263 |
OGFC |
|||||
Depth(cm)
|
Residual ion concentration (mol/L) |
||||
primitive |
1d |
7d |
14d |
21d |
|
0-1 |
0.2604 |
0.2301 |
0.1798 |
0.1141 |
0.0891 |
1-2 |
0.2756 |
0.2631 |
0.2342 |
0.1844 |
0.1467 |
2-3 |
0.2798 |
0.2756 |
0.2604 |
0.2415 |
0.2199 |
3-4 |
0.2812 |
0.2825 |
0.2714 |
0.2466 |
0.2284 |
As could be seen from Table 9, residual ion concentrations in the surface layer (0-1cm) and deep layer (3-4 cm) of three gradation type (AC-10, SMA, OGFC) decreased respectively by 30.47%, 28.56%, 65.8% and 7.52%, 8.39%, 18.76% in 21d. It is proved that the gradation type has some influence on the salt release ability of salt storage asphalt mixture. Since water cannot seep down, the effect on the surface layer is only at the stage of immersion in still water. For better connected gradation types, such as OGFC, water mainly scours the mixture, resulting in the release of the active ingredient at a relatively faster and higher concentration. For the gradation type with interconnected pores, the active component in the deep layer decreases more quickly because the water can be immersed into the deep layer more easily. However, because of the slow process of water seepage, the release process of the active ingredient in the deep layer is slower with the increase of cycle, which can ensure the continuity and slow release of the active ingredient. (line 549-561)
Point 9:The author’s team should present the research needs and suggestions for future studies on Advances in Salt-storage Materials, which problems should be focused on?
Response 9: (1) It is suggested that future research should be carried out in the control of slow release, environmental response and corrosion protection of salt storage materials, so as to achieve long-term snow melting and ice suppression. The differences and accuracy of different evaluation methods are compared and analyzed in order to put forward reasonable test conditions and evaluation criteria. (line 604-609)
- It is suggested that different kinds of materials should be selected and suitable improvement measures should be taken to ensure the durability of the mixture, taking into account road performance, ease of use, and economic factors. Theoretical research on road performance of salt-storage asphalt mixture especially the multi-scale structural damage and performance degradation model under multi-field coupling is strengthened. (line 620-625)
- It is recommended that in the future there be more accurate means of quantitatively evaluating the snow and ice melting effect of salt storage pavements, as well as establishing more accurate functional life prediction models. (line 634-637)
